# Using Insect Larvae and Their Microbiota for Plastic Degradation

**DOI:** 10.3390/insects16020165

**Published:** 2025-02-05

**Authors:** Isabel Vital-Vilchis, Esther Karunakaran

**Affiliations:** School of Chemical, Materials and Biological Engineering, The University of Sheffield, Sheffield S1 3JD, UK; ivv0211@hotmail.com

**Keywords:** plastic degradation, plastivore larvae, waxworms, mealworms, superworms

## Abstract

Plastic pollution represents a serious environmental problem around the world. Less than 10% of plastic made is recycled, and the rest is either incinerated, accumulates in landfills, or is discarded in the natural world, where it becomes a severe health threat for animals and humans. Thus, novel, efficient, and environmentally friendly solutions are urgently needed. In this regard, the most novel scientific breakthrough occurred around 2014 when scientists discovered the incredible ability of some insect larvae to feed on plastic. This review covers all the larvae with this ability reported since then, especially waxworms, mealworms, and superworms, as well as the first adult insect “palstivores”: termites. It also reports on their gut microorganisms and enzymes that contribute to plastic uptake.

## 1. Introduction

Plastic pollution represents one of the major global challenges of this era, and yet it has been reported that only 9% out of the 9 billion tons of plastic that has ever been produced has been recycled [1]. This pollution is a major global environmental threat that could cause serious changes in the equilibrium of every ecosystem; for instance, it can cause practically irreversible changes to the carbon cycle and to other nutrient cycles, as well as changes in the composition of soils, sediments, and aquatic environments [2]. It can also cause a wide range of health issues for both animals and humans that are described in more detail below in this review.

As plastic accumulates in the environment at alarming rates, new and more effective solutions to address this problem are needed. A novel biotechnological approach is to degrade this plastic into its original monomers using microorganisms and their enzymes so that these monomers can potentially be upcycled into new high-value products later [3]. An even newer biotechnological trend is the use of insect larvae for the same purpose [4].

Even though common insect pests, such as *Rhyzopertha dominica* (commonly referred to as the lesser grain borer) and *Tenebroides mauritanicus* (Cadelle beetle), have been observed to penetrate packaging materials since the 1950s, the main concern at the time was to protect packaged food from these invaders [5,6]. In the 2000s, for the first time, a group of students publicly showed mealworms consuming Styrofoam plastic at a science fair [7].

However, the first scientific report suggesting the revolutionizing idea of using insects to fight plastic pollution did not come until 2014 using the larvae *Plodia interpunctella* [8,9]. In this study, scientists observed *P. interpunctella* chewing and eating PE films; they then proceeded to isolate the first gut PE-degrading strains *Enterobacter asburiae* YT1 and *Bacillus* sp. YP1. This report was followed by the first report of full PS mineralization into CO_2_ by *T. molitor* larvae [10] and the first observation of the waxworm (*Galleria mellonella*’s larvae) degrading PE in 2017 [11]. In the same year, this discovery hit the news and was published in *National Geographic* to reach the general public, where Dr. Federica Bertocchini was acknowledged as the discoverer [12]. Later in 2022, she and her research group in Spain identified four novel waxworm saliva enzymes responsible for this degradation and named them Demetra, Cibeles, Ceres, and Cora, which are the first plastic-degrading enzymes ever isolated from an invertebrate organism [13,14]. Some other relevant events also include the introduction of the term “plastivore” to describe insect larvae or any other organism capable of using plastic as carbon feedstock [15] and the report of plastic-degrading yeasts from adult termite guts [16]. All these important events are illustrated in Figure 1 in chronological order.

The number of scientific papers related to plastic-eating larvae has grown every year, and yet the number of papers to date is still low. Till June 2024, only 366 papers resulted from the keyword search “insect larvae to degrade plastic” on PubMed, and only a couple of them are literature reviews. The first literature review ever published that summarizes the insect degradation of plastics was released in the year 2021 [17], and the latest is from 2024 [7,18,19], but there are very few reviews in between [4,20,21]. This review expands the knowledge on plastivore larvae even more and includes the latest research information available to date (till June 2024). Specifically, it is the most comprehensive and thorough literature review about the waxworm (*Galleria’s mellonella* larvae) published to date, but it also reviews mealworms (*Tenebrio molitor*) and superworms (*Zophobas atratus*), focusing on the identification of the plastic-degrading microorganisms that have been identified in these larvae’s gut and on the understanding of the potential molecular mechanisms present in these larvae for degradation to take place. It also describes the latest discoveries, which include the identification of novel enzymes from waxworms’ saliva and the first potential adult plastivores: termites.

## 2. Plastic Pollution

Plastics represent a wide range of synthetic or semisynthetic materials that consist of long chains of repeated units (monomers) [22]. Around 8300 million metric tons of virgin plastics had been produced by 2017 from non-renewable petrochemical feedstocks, and only a small proportion has been recycled or incinerated. It is estimated that if the current trend continues, approximately 1200 metric tons of plastic waste will accumulate in landfills or in the natural environment by 2050. Whilst this analysis includes thermoplastics, thermosets, polyurethanes (PURs), elastomers, coatings, and sealants, it mainly focuses on the most abundant resins and fibers: high-density polyethylene (HDPE); low-density and linear low-density PE (LPE); polypropylene (PP); polystyrene (PS); polyvinylchloride (PVC); poly-ethylene terephthalate (PET); PUR resins; and polyester, poly-amide, and acrylic (PP&A) fibers [23].

The molecular structure of these common plastic resins (81% plastics), along with their density, crystallinity, life span in the environment [24,25], common uses, and demand distribution by resin type in the year 2018 in Europe [26] are reported in Figure 2. The other 19% of resins that is not presented in Figure 2 includes PTFE for cable coatings in communications, PMMA for touch screens, PC for roofs and eye glasses, PBT as optical fiber, ABS for keyboards, LEGO toys, and others [26]. The half-life of PU is still unknown [27].

The life span of plastics in the environment could be reduced in the presence of insect larvae. To mention a couple of examples, one larva of *P. davidis* can ingest ≈ 2.4 mg of PS per day and survives only on this material [28], while *Uloma* can consume 0.37 mg of PS per day per larva [29], and 150 larvae of *Galleria mellonella* are capable of consuming 0.88 g of PE and 1.95 of PS in 21 days [30].

The ubiquitous distribution of plastic contamination in both the terrestrial and marine environments has identified the phenomenon as a key geological indicator of the Anthropocene [31], which is an epoch of time defined by the domination of humanity over surface geological processes [32]. Plastic pollution is a serious issue that affects animal and human life. In the sea, for example, it has been reported that over 260 marine species, including mammals, seabirds, turtles, and invertebrates, can become entangled in or ingest plastic waste, which impairs their movement, feeding, and reproductive capabilities, or causes internal lacerations and ulcers, ultimately resulting in death [33].

One of the major problems of plastic pollution is that the incomplete degradation of plastics in the environment leads to the accumulation of microplastics (particles of less than 5 mm) rather than the complete mineralization of the material [34,35]. In humans, microplastics enter the human body through inhalation, ingestion, and dermal contact, and although more human health hazard studies related to microplastics are needed, some potential hazards include metabolic disorder, inflammation, oxidative stress, and multisystem adverse effects (respiratory and digestive) [36,37], as well as potential male and female fertility issues [38]. It can also induce DNA damage and oxidative stress, which, in turn, lead to carcinogenesis [39]. Unfortunately, this threat is now imminent as microplastics have been found in the marine environment, soil, in drinking water, and even in commonly consumed food like fish, vegetables, sugar, honey, and salt [40]. Some studies also show that these microplastics are indeed present in humans’ blood [41], stool, lungs, placentas, internal organs [42], and in reproductive systems [43].

Some solutions for this problem include the recycling, incineration, or disposal of plastics in designated landfills. Unfortunately, even though plastic recycling has existed for decades, scientists estimate that only 9% is recycled globally, 12% is incinerated, and 79% is either in landfills or in the environment [44]. These numbers are surprising because, in principle, most plastics are recyclable; however, there are many factors that represent a barrier towards recycling. For example, the contamination of these items in the form of labels, food, or other products in recycling bins may inhibit recycling entirely. Some plastic items are a complex bend of chemical additives which are harmful for human health, which makes recycling dangerous for workers, and other items are so unique that they cannot be recycled together [44].

Incineration is a method that can permanently degrade and eliminate plastic waste; nevertheless, the residual ashes from municipal incinerators are still a source of microplastics [45]. Moreover, the process releases toxic volatile organic compounds into the air. These compounds include chlorinated and aromatics such as benzene and chloroform [46]. Consequently, more environmentally friendly solutions are needed.

To sum up, plastics break down into microplastics that are a serious global environmental and public health threat. This is especially true in the cases of PE, PS, PET, PVC, PU, and PP because these are the most abundant types. Solutions to this problem exist already, for example recycling and incineration, but they all have drawbacks and limitations; therefore, more research and effort should be put in the future.

## 3. Degradation of Plastics—A General Perspective

The degradation of plastics takes place because of abiotic and biotic factors present in the environment. It is also common to observe both the factors contributing consecutively when, for example, a photodegraded bottle is attacked by microbes, as shown in Figure 3 [47]. Plastic is considered as being degraded by abiotic factors when there is any change that may cause depolymerization, a change in its physical properties, the alteration of its chemical composition, mass loss, or complete mineralization into carbon dioxide and water [48,49]. When biotic degradation results in fragments or microplastics, this process is considered the bio-disintegration of the plastic, whereas if the plastic is entirely assimilated and mineralized inside the cell, it is considered biodegradation [34].

Overall mass loss is the parameter commonly used to study plastic degradation rates [49]. These rates are hard to estimate because of the wide variety of factors affecting the process in different environments; some plastic life span estimations are presented in Figure 2.

Of all the mechanisms described above, one of the most important mechanisms is photodegradation [34]. In photodegradation, high-energy ultraviolet (UV) irradiation UV-B (290–315 nm) and medium-energy UV-A (315–400 nm) initiate radical-mediated plastic degradation [34,50,51].

### Microbial Degradation of Plastics

One of the most relevant events with regard to the microbial degradation of plastics is the discovery of a new bacterium species, *Ideonella sakaiensis,* in 2016 outside a bottle-recycling facility in Japan [52]. This bacterium breaks down PET using two novel enzymes. The first one was labelled as PETase (NCBI accession number A0A0K8P6T7.1) and converts PET to Bis(2-Hydroxyethyl) terephthalate (BHET), TPA, and mono(2-hydroxyethyl) terephthalic acid (MHET), which, in turn, is converted into more terephthalic acid (TPA) and ethylene glycol (EG) monomers by the MHETase enzyme [53] (NCBI accession number A0A0K8P8E7.1), as shown in Figure 4. After this discovery, dozens of other new PETases have also been identified from several other bacteria. Some examples are as follows: *Vibrio gazogenes*, *Oleispira antarctica*, *Polyangium brachysporum* [54], *Marinobacter* sp. [55], *Ketobacter* sp., and *Thermobifida* [56].

Both the final PET degradation products, terephthalic acid (TPA) and ethylene glycol (EG), are either further metabolized by cells through the Krebs cycle for biomass accumulation, or they are converted into high-value products [57]. TPA is converted, for example, into vanillic acid, muconic acid, catechol, pyrogallol, gallic acid, and adipic acid [57], while ethylene glycol is separated and used mainly to produce polyester fibers and antifreeze products [58,59]. BHET is also used in the industry for making resins, coatings, foams, and tissue scaffolds [60], or can be further hydrolyzed inside the cell into more MHET and TPA by esterase enzymes [61].

The number of research publications reporting plastic-degrading microorganisms keeps increasing every day, and by 2020, approximately 436 different species had been reported [62]. These species include bacteria from the classes *Actinobacteria*, *Firmicutes*, *Cyanobacteria*, *Proteobacteria,* and *Bacteroidetes* [62], while plastic-degrading fungi are found in eleven classes in the fungal phyla Ascomycota (*Dothideomycetes*, *Eurotiomycetes*, *Leotiomycetes*, *Saccharomycetes*, and *Sordariomycetes*), Basidiomycota (*Agaricomycetes*, *Microbotryomycetes*, *Tremellomycetes*, *Tritirachiomycetes*, and *Ustilaginomy-cetes*), and Mucoromycota (*Mucoromycetes*) [63].

To name a few, bacteria such as *Cupriavidus necator* H16 [64], *Pseudomonas putida* LS46, and *Pseudomonas putida* IRN22 have also been discovered to degrade polyethylene [65], while *Pseudomonas putida* CA-3 can be fed styrene to accumulate intracellular polyhydroxyalkanoates [66]. A unique *Raoultella* sp. DY2415 strain from petroleum-contaminated soil can degrade PE and PS film [67], while the fungi *Aspergillus fumigatus* and *Phanerochaete chrysosporium* degrade a wide range of plastics [62]. The countries that have isolated the most strains are Japan (14.1%) and India (13.8%) [62].

In an effort to compile all this new information, in 2022, the database PlaticDB was created “https://plasticdb.org/ (accessed on 24 January 2025)”. To this day, the database contains 753 organisms and 219 proteins that include cutinases, esterases, PETases, etc. [68]. Cutinases, specifically, are hydrolases that degrade cutin, which is a component of higher plant cuticles, and they have been extensively studied to degrade plastics (PET, PE, PU, Poly (butylene succinate) (PBS), and Poly (ε-caprolactone) (PCL)) [69]. They are usually isolated from thermophilic actinomycetes such as *Thermobifida fusca* (*KEGG*: Tfu_0882) [70]. Interestingly, it is also possible to discover new plastic-degrading enzymes using metagenomics from a mixed-cultured sample rather than an isolated microorganism. This is the case for *Tm*Fae-PETase discovered by Mamtimin, T., et al. [71] from mealworms’ frass [71]. Zrimec, J., et al. [72] also conducted the metagenomics analysis of environmental global samples from oceans and soils to compile a wide catalogue of over 30,000 nonredundant enzyme homologues with the potential to degrade 10 different plastic types.

However, despite the number of microorganisms and enzymes available, most of them have low activity levels and are not thermostable [52,73]. As a result, efforts have been made to engineer these proteins to increase activity and thermostability. Some examples of these enhanced proteins are ThermoPETase, HotPETase [74,75], DuraPETase [76], and the novel FAST-PETase (FAST-PETase: functional, active, stable, and tolerant PETase) [77] from *Ideonella sakaiensis.* Moreover, for a more environmentally friendly approach, native *I. sakaiensis* PETase has also been successfully expressed in the chloroplast of the microalgae *Chlamydomonas reinhardtii* [78].

In conclusion, plastics degrade in the environment over time as a consequence of several abiotic factors such as temperature and humidity. They also degrade thanks to the presence of enzymes from a great variety of plastic-eating microorganisms such as *I. sakaiensis.* These cells could be the key to not only fighting plastic pollution, but to obtaining high-valuable products from this plastic.

## 4. Insect Plastic Degradation—Order: Lepidoptera (Butterflies and Moths)

Lepidoptera is an order of winged insects, and it is the second largest order there is, with approximately 180,000 described species [79]. Aside from the wings, the more representative features are the presence of scales and the proboscis (tubular sucking organ) [80]. The larvae of the following insects from this order have been reported to have plastic-degrading capabilities, A. grisella, P. interpunctella, C. cephalonica, S. *frugiperda,* and *Galleria mellonella,* from which this last one is by far the most commonly studied (refer to Figure 5).

Interestingly, similar to *Galleria mellonella*, the larvae from *Achroia grisella* and from the beetle *Uloma* feed on long-chain hydrocarbon beeswax and can degrade the plastics PE and PS (pre-print study) [87]. The larvae from *Plodia interpunctella* also eat both beeswax and PE [8]. The positive relationship between the capability of eating beeswax and the capability of eating plastic might not be a coincidental one, but rather a case of cause and effect since it has been suggested that similar metabolic approaches are used to degrade both these compounds [88].

### The Waxworm Galleria mellonella (Fabricius, 1798) [Lepidoptera: Pyralidae] Degrades Plastic

Commonly referred as the greater wax moth, *Galleria mellonella* is a natural honeycomb pest that has contributed to the decline of bee populations at a global scale due to the larvae’s capability to feed on wax [86]. In science, these larvae’s importance has gradually increased as a model organism for biomedical studies [89]. They are specially used as an infectious-disease model due to the presence of an immune system that is similar to that of vertebrates [90].

To date (June 2024), the PubMed entry “*Galleria mellonella* to degrade plastic” gives 43 entries, from which only 28 are related to the larvae’s capacity to degrade plastic, and these are summarized here.

The degradation of plastics using the larvae from *Galleria mellonella* (commonly referred as waxworm) is a fairly novel research topic. The first experiment reporting the capability of this insect to degrade polyethylene (PE) was presented in 2017, when Bombelli, P., et al. [11] left worms in a polyethylene bag and observed that they were eating it. The plastic degradation capability of *Galleria mellonella* has also been found to apply to other petroleum-based plastics, such as expanded polystyrene and polypropylene [91] and for bio-plastic polylactic acid (PLA) [92]. *Galleria mellonella* is naturally capable of decomposing long-chain hydrocarbons from beeswax without the help of intestinal microorganisms using specific carboxylesterases, lipases, and fatty-acid metabolism related enzymes. It has been hypothesized that a similar metabolic approach is used to degrade plastic by the waxworm [88]. However, plastic is not nutritious enough, as studies show that most larvae (≥50%) living on an exclusive PE diet lose weight and die in between 3 and 15 days, indicating that a supplementary diet is necessary [30,93,94], or the use of older larvae (last developmental stage 25–30 mm) [95] for this type of plastic bioremediation to take place. Pre-treating low-density polyethylene under solar radiation for 15 days before feeding the larvae with the material is also being suggested as another technique to increase the plastic degradation rate and the larvae’s survival [96].

Polyethylene degradation starts with the expression of salivary enzymes after exposing the larvae to the material [97]. Sanluis-Verdes, A., et al. [13] discovered and published the first report of two novel enzymes isolated from waxworm saliva with the capability of oxidizing and depolymerizing polyethylene (PE) after only a few hours of exposure to the material at room temperature and a neutral pH. These enzymes, named Demetra and Ceres, are classified as arylphorin and hexamerin, respectively. Gas Chromatography–Mass Spectrometry (GC-MS) was used to confirm the presence of degradation products, such as small, oxidized aliphatic chains in the PE treated with saliva. This discovery opened the door to new ground-breaking solutions for plastic waste management. Unfortunately, it is important to mention that a later study published in *Nature Communications* stated that the plastic-degrading activity of Ceres could not be replicated in their lab and suggests that the original results may have been misinterpreted [98]. A closely related protein (81% shared sequence identity with Demetra) was also described in the original study and named Cibeles. Cibeles forms a heterocomplex with Demetra, but has not proven to degrade PE on its own either [13]. A fourth PE-degrading protein was later identified and named Cora late in 2023 [14]. The 3D structures of all these proteins have been elucidated [14].

Microplastic and plastic depolymerization products are swallowed and further processed in the gut, where the microbiome plays a key role in plastic degradation [20]. *Desulfovibrio vulgaris*, *Enterobacter* sp. D1 [91,99], *Acinetobacter* [15], the fungus *Aspergillus flavus* PEDX3 [100], and the fungus *Cladosporium halotolerans* [101] are examples of microorganisms isolated from waxworm guts with the reported capability of degrading PE in experiments in vitro. For the case of *A. flavus*, for example, microplastics of HDPE were degraded into microplastics with a lower molecular weight when exposed to fungi in liquid culture for 30 days. Chemical changes in the microplastic particles, such as the appearance of hydroxyl, carbonyl, and ether groups, also validate degradation. Two laccase-like, multicopper oxidase enzymes are believed to be responsible for this degradation [100]. Highly similar results were observed when the fungus *Cladosporium halotolerans* was cultivated in an HDPE microparticle suspension [101].

Likewise, the bacteria *Lysinibacillus fusiformis*, *Bacillus aryabhattai*, and *Microbacterium oxydans* isolated from a whole worm body extract are able to degrade and grow using low-density polyethylene LDPE as a carbon source [65].

Aditionally, up, other microorganisms from this worm have been studied and proven to be capable of acting on other plastics different from polyethylene (PE). For example, *Bacillus cereus* can degrade polypropylene (PP) in vitro [102], while the mastication of expanded polystyrene (EPS) and polypropylene (PP) increase the abundance of *Enterococcus* sp. in the gut [91]. Also, the genera *Bacillus* and *Serratia* and the bacterium identified as *Massilia* sp. FS1903 have been associated with polystyrene (PS) degradation [30,103]. Some enzymes, pathways, and gut microorganisms mentioned in this section used by waxworms to degrade plastic are summarized in Figure 6 and Figure 7.

Despite of all the above studies, plastic degradation in the gut cannot be solely attributed to microbiota presence. Gut RNA sequencing and biochemical approaches showed that polyethylene-fed larvae show enhanced fatty acid metabolism (FAM) [104,105]. Additionally, early in 2024, an improved version of the whole genome of *Galleria mellonella* was published (GenBank: JAPDED000000000.1) [106]. In this study, various new, putative, probable PE-degrading enzymes found are highlighted.

As for the case of polystyrene metabolism, a list of possible styrene-degrading enzymes present in the waxworm have been published. Two potential metabolic pathways have been proposed [107,108]: The styrene oxide–phenylacetaldehyde [109] pathway is also expressed in the presence of beeswax [110] (refer to Figure 7A). However, this pathway has never been scientifically confirmed, as there is no study reporting the presence of styrene as a digestion product in either microbes or insects.

**Figure 7 insects-16-00165-f007:**
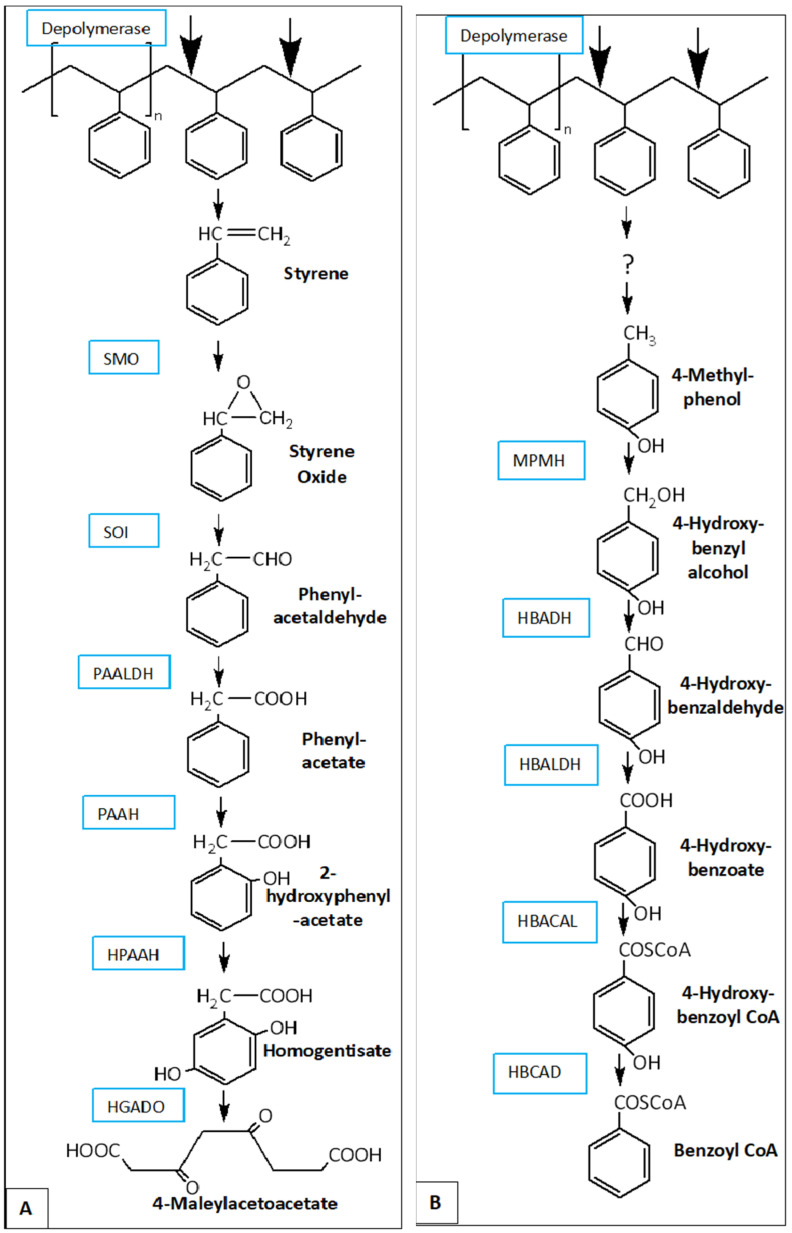
Proposed metabolic pathways for PS degradation in *G. mellonella* [108]. (**A**) The styrene oxide–phenylacetaldehyde pathway (**B**) The 4-methylphenol–4- hydroxybenzaldehyde–4-hydroxybenzoate pathway. SMO: styrene monooxygenase; SOI: styrene oxide isomerase; PAALDH: phenacetaldehyde dehydrogenase; PAAH: phenylacetate hydroxylase; HPAAH: 2-hydroxyphenylacetate hydroxylase; HGADO: homogentisate 1,2-dioxygenase; ?: Unknown intermediate; MPMH: 4-methylphenol methyl hydroxylase; HBADH: 4-hydroxybenzyl alcohol dehydrogenase; HBALDH: 4-hydroxybenzaldehyde dehydrogenase; HBACAL: 4-hydroxybenzoic acid-CoA ligase; HBCAD: 4-hydroxybenzoyl-CoA reductase [108].

The other proposed pathway is the 4-methylphenol–4- hydroxybenzaldehyde–4-hydroxybenzoate pathway (Figure 7B), which is used prior to the β-oxidation pathway (main FAM process) in the larvae’s intestines [108].

In closing, several lepidopterans larvae have been reported to consume different types of plastic, and some examples are larvae from the lesser wax moth, the Indian meal moth, the rice moth, and the fall armyworm, yet the most commonly studied by far is the waxworm. This last worm uses a set of newly discovered saliva enzymes, followed by a wide range of gut microbiota (Ex. *Enteroccocus*, *Bacillus*, and *Massilia*) and their own metabolic pathways (fatty acid-related pathways) to convert plastic (PE, PS, and PP) into biomass.

## 5. Insect Plastic Degradation—Order: Coleoptera (Beetles and Weevils) [111]

The order Coleoptera represents the largest group of insects, with 40% of the known insect species. In this group, generally, the wings develop internally, but some have no wings [111]. Plastivore Coleoptera larvae include the larvae from the beetles *Alphitobius diaperinus*, *Plesiophthalmus davidis*, *Tribolium castaneum*, *Uloma*, *Tenebio molitor*, *Tenebrio obscurus,* and *Zophobas atratus*. The types of plastic and microbiome associated with these insects are shown in Figure 8. Additionally, a more recent study also added the soil-dwelling grub larvae of the beetle *Protaetia brevitarsis* to the list of polystyrene (PS) consumers [112].

### 5.1. The Yellow Mealworm Tenebrio molitor (Linnaeus, 1758) [Coleoptera: Tenebrionidae] Degrades Plastic

Commonly referred to as mealworms, *Tenebrio molitor* larvae are commonly used as a protein source for domestic animals (monogastric animal feed) [121], for fish [122] and can also be grown for human consumption [123,124]. In science, it has been studied as a model for cellular and humoral immunity against pathogenic infections [125]. A representation of an adult beetle is shown in Figure 9.

Starting from 2015 to date (June 2024), the search in PubMed “*Tenebrio* to degrade plastic” gives 82 entries, from which 67 are related to the larvae’s capacity to degrade plastic. Most of those papers were used for this review.

Mealworms, at the moment, are growing a reputation as polystyrene (PS) plastic eaters [10,30,105,117,126,127,128,129,130,131]. These larvae are capable of converting ≈47% of ingested Styrofoam (a common PS product) into CO_2_, and the residue (≈49%) is excreted as fecula, with a limited fraction incorporated into biomass [10]. Although *Galleria mellonella* can degrade PS [132], mealworms lose their capacity to degrade PS plastic when gut bacteria are inhibited [130,133], which suggests a stronger dependency on the microbiota to degrade PS. Even so, it has also been demonstrated that the mealworm secretes emulsifying factors that increase plastic bioavailability in the gut [128], as well as a wide range of oxidases, cytochrome P450, monooxygenases, superoxidases, and dehydrogenases, and other enzymes related to fatty acid metabolism [134,135].

Mealworms of approximately 3–4 instars (20–25 mm in length) fed only PS are able to survive and complete their entire life cycle and grow into adult beetles [10,131,135]. This can partially be explained by the fact that the gut microbiome, specially with the genus *Klebsiella*, is capable of nitrogen fixation, thus the worm is supplied with this element as well [136]. Still, it is recommended to supplement with corn flour (*T. obscurus*) or wheat bran (*T. molitor*), sucrose, and hydrate with H_2_O [137] to increase the PS degradation rate and enable the breeding of a second generation with favorable capabilities for PS degradation as well [129,131,138]. 

It has been hypothesized that the mechanism used to degrade PS is similar to the mechanism described for *Galleria mellonella* in Figure 7A, in which PS is degraded into styrene first [105], and after several intermediate steps, the benzene ring is destroyed [132] and assimilated through the β-oxidation pathway [108].

Additionally, *T. molitor* larvae can also biodegrade polyethylene (PE) through the mechanism presented in Figure 9 [127,139,140]. They can also degrade PP, PVC [116,141,142], Nylon 11 Polymer [143,144], Polyethylene terephthalate (PET) [134], melamine formaldehyde (MF) [145], and the biopolymer PLA, but the mechanism for these polymers remains unknown [146]. *T. molitor* can even chew and ingest polyurethane (PU), but the digestion/degradation of this plastic has not been demonstrated [147,148,149]. During the COVID-19 pandemic, polypropylene (PP) face mask production and contamination increased considerably, and it was observed that *T. molitor* can consume face masks [150]. The capability of biodegradation can be affected by the molecular weight, branching, and crystallinity of this material [151].

The mealworm’s gut bacterium, *Exiguobacterium* sp. strain YT2, degrades polystyrene (PS) [130]. *Citrobacter* sp. and *Kosakonia* sp. were also found in the gut and strongly relate with PE and PS consumption [127], as well as the bacterium *Priestia megaterium* S1 [152]. The other strains related to PS degradation include *Erwinia olea*, *Lactococcus lactis*, *Lactococcus garviae* [137], *Serratia marcescens*, *Pseudomonas aeruginosa*, *Acinetobacter septicus*, *Agrobacterium tumefaciens*, *Klebsiella grimontii*, *Pseudomonas multiresinivorans*, *Pseudomonas nitroreducens*, *Pseudomonas plecoglossicida*, and *Yokenella regensburgei* [117,126,128,153]. Bacteria from the family *Enterobacteriaceae*, such as *Enterobacter hormaechei LG3* [154], and the families *Spiroplasmataceae*, *Enterococcaceae* [129,132], *Staphylococcus,* and *Rhodococcus* [135] also play a role in PS degradation. It is important to note that a study where mealworms from three different regions in China were compared showed that the larvae from different regions have different metabolisms [155], which suggests that the gut microbiota can change depending on the environment of the larvae, diet, and even depending on the PS molecular weight provided [156], yet PS consumption is ubiquitous to this species [157].

Additionally, the family *Enterobacteriaceae* has also been linked with polyurethane (PU) degradation, along with the family *Hafnia* [158], while the genera *Spiroplasma*, *Dysgonomonas*, and *Hafnia-Obesumbacterium* are associated with PET degradation [134]. The gut bacteria from *Tenebrio molitor* are even capable of degrading vulcanized poly(cis-1,4-isoprene) rubber (vPR) (strain *Acinetobacter* sp. BIT-H3) [159]. They are also capable of degrading PVC and PP, but the genus of these microorganisms has not been elucidated [141,142]. *Tenebrio molitor*’s microbiome in relation to plastic-degradation is illustrated in Figure 10.

Interestingly enough, although less commonly studied, a comparison between the yellow mealworms (*T. molitor*) and dark ones (*T. obscurus*) shows that the latter degrade PS at higher rates [129]. *T. obscurus* larvae also degrade LDPE using gut bacteria mainly from the genera *Spiroplasma* and *Enterococcus* [118,119].

### 5.2. The Superworm Zophobas atratus (Fabricius, 1776) [Coleoptera: Tenebrionidae] Degrades Plastic

*Zophobas morio* (Fabricius, 1776) is a dark beetle [Coleoptera: Tenebrionidae] that is currently considered as being the same species as *Zophobas atratus*. It was also previously identified as *Tenebrio morio* and/or *Helops morio* and commonly referred as the giant mealworm beetle, which has been the cause of confusion and controversy [160]. *Zophobas* larvae are commonly refer as superworms [160]. These larvae are highly nutritious and are promising as fish, poultry, and pig feed [160].

To date (June 2024), the search in PubMed “*Zophobas atratus* to degrade plastic” gives 17 entries, from which all 17 are related to the larvae’s capacity to degrade plastic. All 17 studies were used for this review.

*Zophobas* is also a main polystyrene (PS) consumer. A comparison in a 30-day-long experiment between the larvae of *Tenebrio molitor* (yellow mealworm), *Galleria mellonella* (greater wax moth), and *Zophobas atratus* (superworm) showed that the latter have the strongest polystyrene consumption capacity and the highest survival rate of the three [132], being able to consume four times more PS than the yellow mealworm per day [161]. Superworms also outdid the yellow mealworms by 11 folds on PU consumption in another study [158]. But similarly to *Tenebrio*, this capability is lost when the gut microbiota are suppressed using antibiotics [161].

Moreover, new research indicates that the superworm’s microbiota are also capable of degrading PE, PP, PVC, and PET [120,162,163,164], and even polyurethane (PU) [165], melamine formaldehyde (MF) [145], ethylene vinyl acetate (EVA) [166], and polybutylene succinate (PBS) [167].

PE and PS degradability is being attributed to *Pseudomonas aeruginosa* [120,168] and *Enterococcus* (also associated with PU degradation) [165] while *Citrobacter* is associated with PE and PVC [163]. *Brevibacterium* [169], *Dysgonomonas* and *Sphingobacterium* are associated with PS, and *Mangrovibacter* with PU degradation [165]. (summarized in Figure 11).

Little is known about the mechanisms these specific larvae use to degrade plastic, but PS degradation seems to be partially achieved by the synergistic effect of the generation of reactive oxygen species (ROS) inside the gut and the production of oxidases and other enzymes by the microbiome [170].

In general, it can be said that out of the large insect order Coleoptera, yellow mealworms and superworms stand out for their capacity to eat plastic. They are specially known for eating polystyrene, but they can eat other plastics too such as PE and PVC. One interesting difference between these Coleopterans and waxworms is that they are much more dependent in their microbiota for plastic degradation; in fact, they lose their plastic-eating capacity when the microbiota are inhibited with antibiotics.

## 6. Insect Plastic Degradation—Order: Blattodea (Cockroaches and Termites) [18,87,88,171,172]

The list of plastivores in this review includes all the insects mentioned in the previews reviews [18,172] from the orders Lepidoptera and Coleoptera and expands to include the order Blattodea, which could be further explored in the future for plastic degradation

The Order Blattodea consists of two main insect groups, cockroaches and termites, and both these groups have shown plastivore abilities. Cockroaches have been observed eating plastic films since the 1950s [6]. In a recent study from Li, M.-X., et al. [173], the cockroach *Blaptica dubia* (Seville, 1983) [Blattodea: Blaberidae] was able to digest up to 46.6% of ingested PS within 24 h.

Termites have a physical appearance that is similar to that of ants, as observed in Figure 12. They used to be classified in their own order named Isoptera. However, new studies show that they are actually closely related to cockroaches and should be classified in the same order Blattodea [171]. The most characteristic feature of these insects is that they feed on wood, which is composed mainly of polymers of cellulose, hemicellulose, and lignin [174,175]. Lignocellulose and plastic polymers have similar physicochemical features; for example, their carbon chains have similar chemical bonds and hydrophobicity properties, which has led to the belief that termites could also be plastivore organisms [176]. The relationship between lignocellulose and plastic consumption is further supported by the fact that, as mentioned before, some enzymes like cutinases, which are involved in degrading the cutin present in plant cuticles, are very well documented plastic degraders [69,70]. Another example is the novel feruloyl esterase-like enzyme named *Tm*Fae-PETase by Mamtimin, T., et al. [71], which is a lignocellulose-degrading enzyme present in *T. molitor.*

In fact, higher adult termites (*Nasutitermes nigriceps*) have been observed and reported degrading wood–HDPE plastic composites (WPCs) in one study from Yucatan, Mexico [177]. In a later study, a group of three previously isolated yeast symbionts from the guts of adult *Coptotermes formosanus* (termite) showed low-density PE degrading capability and conversion into alkanes, aldehydes, ethanol and fatty acids [16,178].

Some other wood-eating insect examples include the pest *Chrysobothris* sp. which is a beetle that attacks cedar trees [179]; the emerald ash borer (*Agrilus planipennis*), which attacks ash trees [180]; the red bay ambrosia beetle (*Xyleborus glabratus*) that attacks laurel trees [181]; and the Asian long horned beetle (*Anoplophora glabripennis*) that attacks maple trees [182]; as well as other woodboring beetles [183].

From all the information given above, it can be concluded that a great variety of insect species from the order Blattodea, including all types of termite and cockroach, could hypothetically eat plastic as well, but this hypothesis is yet to be tested and proved scientifically. Blattodea insects are the first example of adult insects to be seen degrading plastic, as all previously reported degradation has been reported for larvae only.

## 7. Other Orders from the Class Insecta That Degrade Plastic

The number of plastivore insects is growing every day, and in the latest 2024 review [7], the authors suggested a large variety of insects with potential plastic-degrading capabilities, which include insects from the orders Diptera (example: Black soldier fly), Blattodea (several type of cockroach), and Orthoptera (Ex. the cricket *Gryllodes sigillatus*), as well as other families within the already studied orders Coleoptera (examples: the lesser grain borer or *Rhyzopertha dominica*, the rice weevil or *Sitophilus oryzae*, and the cigar beetle or *Lasioderma serricorne*) and Lepidoptera (The larvae of *Hofmannophila pseudospretella* or the Brown House moth). However, scientific studies are needed to confirm whether there is actual degradation by these species and which microorganisms/enzymes could be responsible. Another recent review estimated that over 23 species of insects (including the 12 insects described in this review) have been observed consuming plastics [184], and this list could expand even further in the future. If the positive relationship between wax degradation and plastic degradation is confirmed, then other wax-eating larvae could be discovered. An example of this type of larva would be the American waxworm *Vitula edmandsii,* which is a honeybee comb pest [185]. Similarly, other woodboring beetles or any other xylophagous (wood diet) larva or insect could be a good candidate for research.

All the above evidence shows that the insect-screening process for identification, testing, and the exploitation of plastivores is far from over, and that this research topic is possibly going to become a major branch of entomological research in the near future.

## 8. Analysis of Plastic Degradation After Exposure to Insect Larvae

If plastic degradation is being performed by the microbiota, then the first step usually consists of isolating the colonies to obtain a pure culture. In order to isolate the plastic-degrading microorganisms, the fecal matter collected has to be diluted and plated in tryptic soy agar (TSA) and in a defined medium as described in the literature [144]. To confirm the capability of these colonies to degrade plastic, several tests can be performed, such as the clear zone assay on an agar plate and turbidity measurements in liquid culture [152]. However, microorganism isolation is not indispensable, and larvae frass can be collected and analyzed directly [127].

Several techniques and protocols are available for the characterization of degraded plastic after exposure to larvae and their microorganisms. The most common approaches are mass loss, physical alteration, chemical structure changes, and the identification of biodegraded intermediates and products [7,186].

Mass loss is the simplest one and consists of measuring plastic weight loss over time [81]. Once full digestion has taken place, the molecular weight of the residual plastic present in frass can be measured using gel permeation chromatography (HT-GPC) [127].

Physical alterations in the plastic can be analyzed by, for example, inspecting for changes in surface morphology using scanning electron microscopy (SEM) [28] or other types of microscopy, such as TEM, AFM, and EFM [187]. Thermal changes in the material can also be measured using thermal gravimetric analysis (TGA) [127].

The particular method needed to analyze chemical structure changes will depend on the plastic polymer type and its recalcitrance [187]. The common chemical changes observed are oxidation reactions that can be analyzed using X-Ray photoelectron spectroscopy (XPS) [28], Fourier-Transform Infrared Spectroscopy (FTIR), Nuclear Magnetic Resonance (NMR), and Energy-Dispersive Spectroscopy (EDS) [187].

Additionally, plastic carbons atoms can also be tracked throughout the entire metabolic process all the way to conversion into biomass and CO_2_ using radioactive isotopes (Ex. radioactive isotope ^14^C, stable isotope ^13^C, or isotopic signature δ^13^C) [187].

Lastly, Gas Chromatography–Mass Spectrometry (GC-MS), which is an analytical technique used to identify and quantify compounds, can be used to confirm the presence of degradation products in liquid culture, such as small oxidized aliphatic chains [13]. The other methods available to identify degradation products are NMR, FTIR, High-Performance Liquid Chromatography (HPLC), and Pyrolysis Gas Chromatography [127,188,189].

To sum up, there is a wide range of equipment and protocols available to effectively prove and analyze plastic degradation by insects. Most studies use more than one method for improved certainty and accuracy.

## 9. Challenges and Future Perspectives

Even though plastivore insects are a new, exciting avenue for the bioremediation of plastic pollution, several challenges need to be overcome before this technology can be industrialized.

One of these challenges is to provide optimal, standardized conditions for larval rearing at the industrial scale to ensure reproducible results in terms of larvae quality (weight, survival rate lipid content, etc.) and the plastic degradation rate. The environmental conditions, such as light exposure, temperature, and ventilation, greatly affect the development of larvae. In the case of *Galleria mellonella*, for example, constant exposure to light significantly reduces their size and delays metamorphosis, so they need to be grown in darkness at a temperature of 28–32 °C. Providing ventilation is also highly important not only to provide oxygen, but to prevent infection too [190]. Therefore, for the use of live larvae for plastic degradation, we consider it essential to have a contained and controlled area (a dark greenhouse for example) with controlled conditions. The other indispensable benefit of the use of an enclosed area is the responsible containment of insects that are recognized as pests.

Another problematic source of variability at the industrial scale is the chemical composition and properties of the waste plastic used as feedstock. The evidence shows that the presence of other contaminants in plastic pollution, such as plastic additives, as well as other factors, may affect the larvae’s digestion [191]. This challenge could be overcome by processing the plastic waste prior to feeding the larvae as it is normally processed for plastic recycling as follows: (1) Sorting and categorizing—In this step, several types of plastic need to be separated from each other. (2) Washing—The impurities that may be toxic for the larvae are removed. (3) Shredding—The plastic is broken down into much smaller pieces. (4) Testing—At this point, the plastic pieces are tested for their quality and density [192]. Additionally, other novel steps shall be explored, for example, (as mentioned previously) pre-treating PE under solar radiation for 15 days before feeding the larvae increases the plastic degradation rate and the larvae’s survival [96].

Concerns have also been raised about the economic feasibility of this technology due to the high cost of breeding larvae and the lack of sufficient research to obtain high-value end products. It has been calculated that it would cost more than EUR 300 and approximately 38 days to degrade 1 ton of low-density LDPE plastic using ≥4 tons of waxworms or mealworms [193], while recycling 1 ton of LDPE costs less than EUR 250 in less time [194]. This problem is also been observed using other types of larva; for example, during the COVID pandemic it was calculated that it would take approximately 100 mealworms 138 days to consume one face mask [195].

In this regard, due to their high fat content, the waxworm, the yellow mealworm, and even the PE plastivore larvae from *Corcyra cephalonica* (up to 60%, 38%, and 43.3% of body weight respectively) could potentially be used for biodiesel production [196,197]. Another solution suggested is the extraction of chitin from adult plastic-fed *Tenebrio molitor* exoskeletons [198], which can then be processed to use as biomedical materials, food additives, cosmetic ingredients, agricultural materials, analytical reagents, and others [199]. Larvae could also potentially be used as animal feed as some studies show that there are no microplastic nor nanoplastic residues present in frass as a result of plastic consumption by larvae [140]. However, more studies are needed to corroborate safety.

On the other hand, it is also important to note that this same study [193] calculated that the process used to degrade 1 ton of plastic using larvae would also release ≥4 tons of CO_2_ into the atmosphere, which is a much larger number than the ≈2.9 tons of CO_2_ that would be produced during plastic incineration [200]. This information is worrying and leads us to wonder if this technology can be used as a sustainable process. In the future, the process will need a more comprehensive Life Cycle Assessment [201] and the possible co-implementation of a CO_2_ capture system.

Lastly, scale-up standardization and the production of high-value end products could be achieved using other, more advanced biotechnological approaches, such as the recombinant expression of insect-derived novel enzymes, or the use of gut-isolated microorganisms to degrade plastics in cell culture. Insect cells specifically are already successfully used as factories for the biomanufacturing of several proteins, vaccines, and vectors for gene therapy [202]. Moreover, the industrial cultivation of microorganisms to obtain biotechnological products has been practiced for thousands of years, starting with the production of wine, beer, and bread [203]. Cell culture plastic degradation would allow for the recovery of plastic monomers that can be converted into high-value components. For example, as mentioned above, TPA monomers from PET degradation can be transformed into vanillin [204], which is considered the second most important flavoring agent after saffron and has a wide variety of applications in the food and beverage industry, but also in the pharmaceutical industry and for the production of home-use products, such as perfumes and deodorants [205].

## Figures and Tables

**Figure 1 insects-16-00165-f001:**
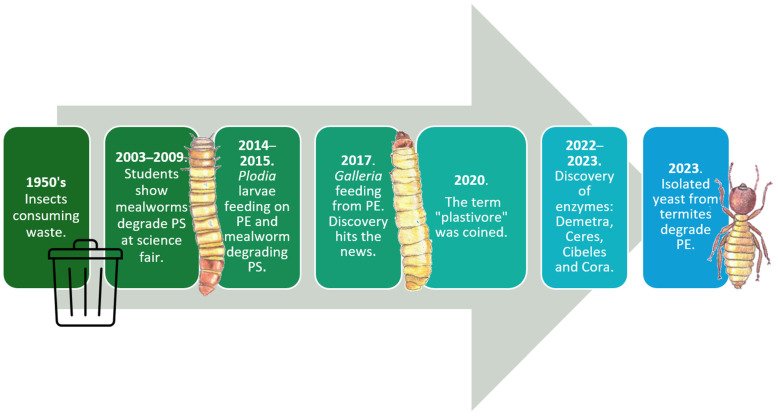
Main historic events related to insects degrading petroleum-based plastic.

**Figure 2 insects-16-00165-f002:**
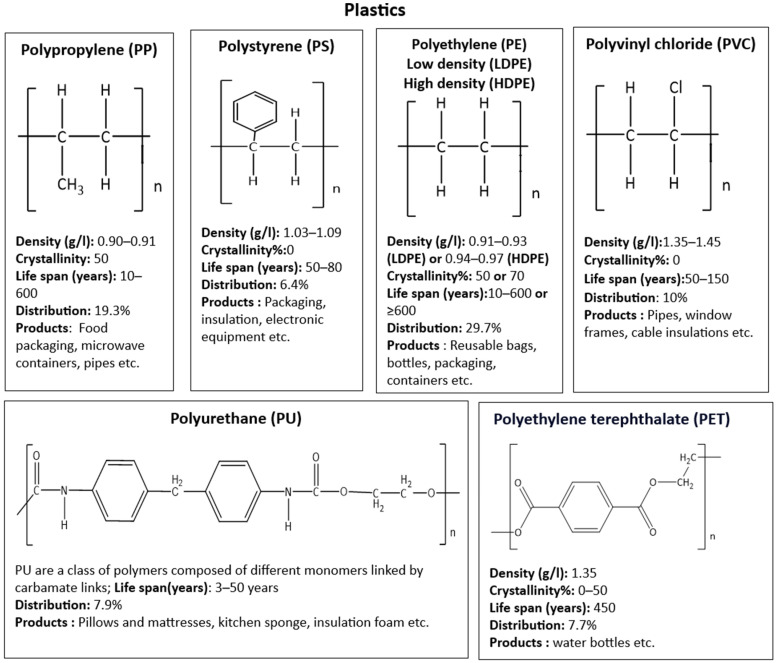
Structure, density, crystallinity, life span, and common uses of most abundant plastic resins [24,25,27].

**Figure 3 insects-16-00165-f003:**
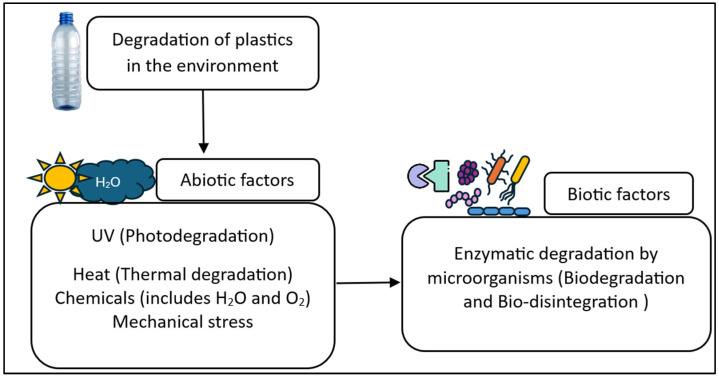
Types of plastic degradation factors in environment [34].

**Figure 4 insects-16-00165-f004:**
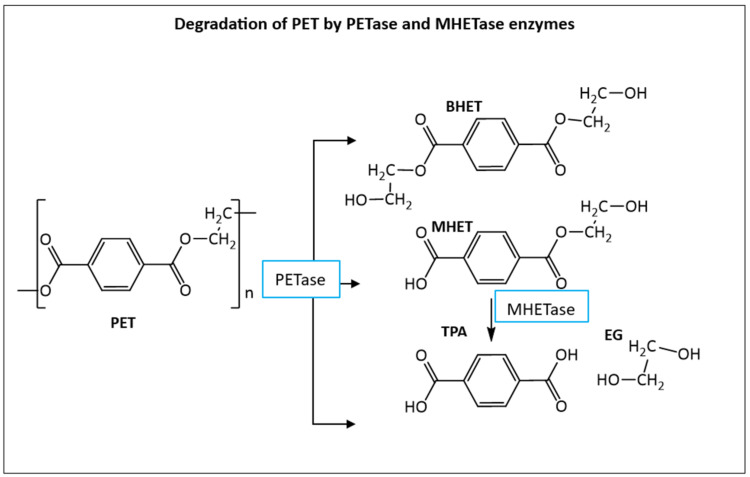
PETase enzyme degrades PET into Bis(2-Hydroxyethyl) terephthalate (BHET), mono(2-hydroxyethyl) terephthalic acid (MHET), and terephthalic acid (TPA). MHETase enzyme further degrades MHET into more TPA and ethylene glycol (EG).

**Figure 5 insects-16-00165-f005:**
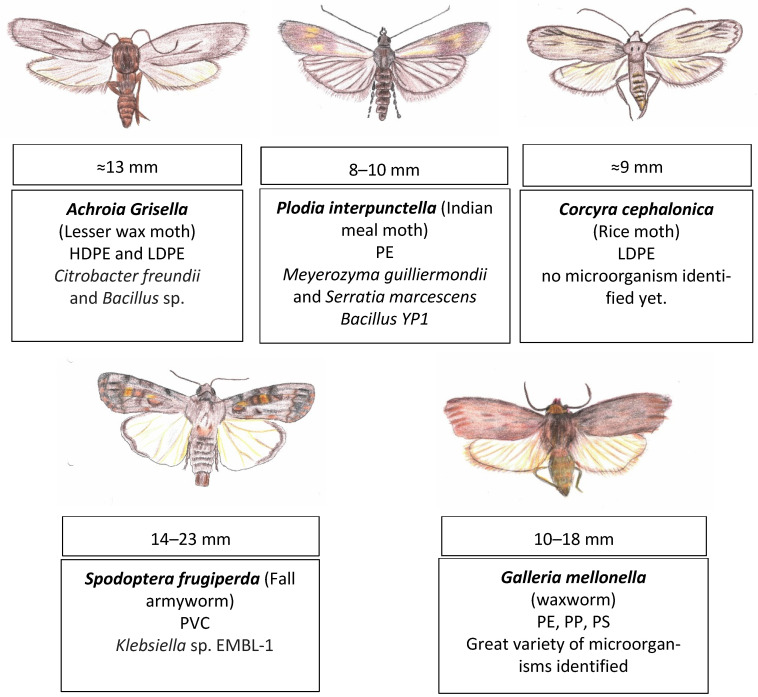
Insects from the order Lepidoptera, whose larvae have been reported to have plastic-degrading capabilities. *Achroia Grisella* [81,82], *Plodia interpunctella* [8,9,83], *Corcyra cephalonica* [84], *Spodoptera frugiperda* [85], and *Galleria mellonella* [11,86]. The body length of an adult is indicated, as well as the larvae’s common name, the plastic degraded, and the microorganism associated with it.

**Figure 6 insects-16-00165-f006:**
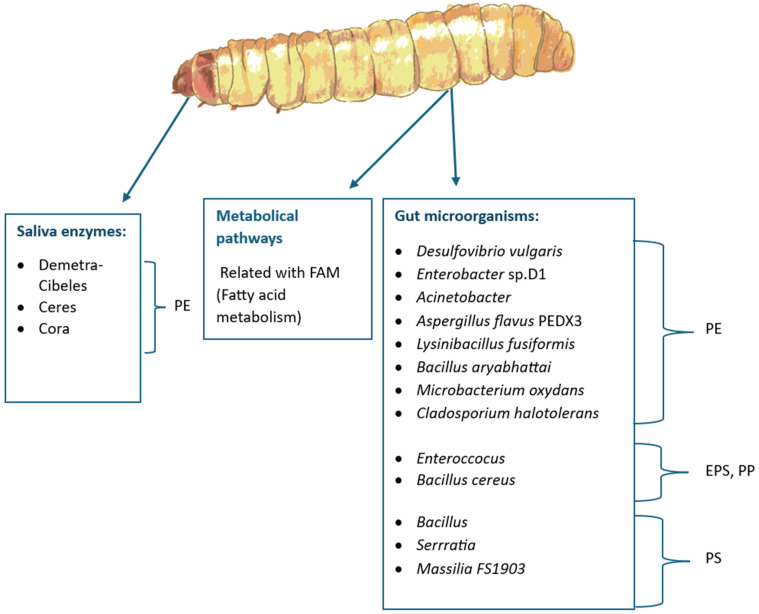
The greater waxworm uses different biological tools to degrade plastic: saliva enzymes, metabolic pathways, and gut microorganisms.

**Figure 8 insects-16-00165-f008:**
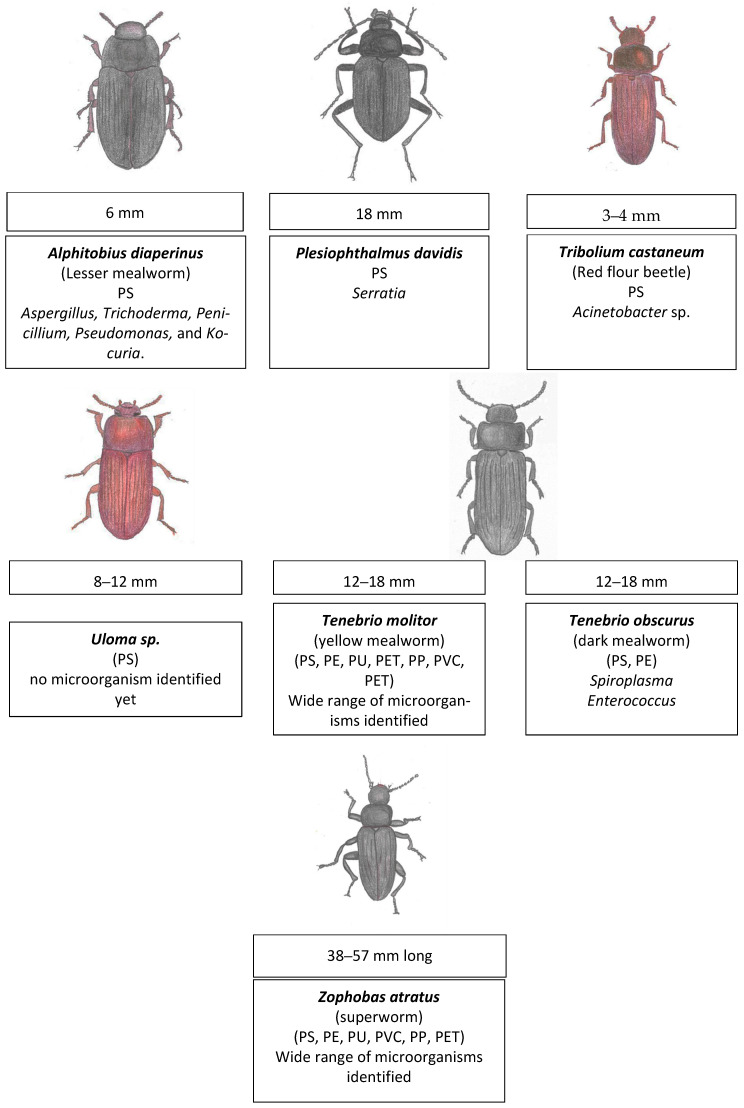
The insects from the order Coleoptera whose larvae have been reported to have plastic-degrading capabilities. *Alphitobius diaperinus* [113], *Plesiophthalmus davidis* [28], *Tribolium castaneum* [114], *Uloma* sp. [29], *Tenebrio molitor* [115,116,117], *Tenebrio obscurus* [118,119], and *Zophobas atratus* [120]. The adult’s body length is indicated, as well as the larvae’s common name, the plastic degraded, and the microorganisms associated with it.

**Figure 9 insects-16-00165-f009:**
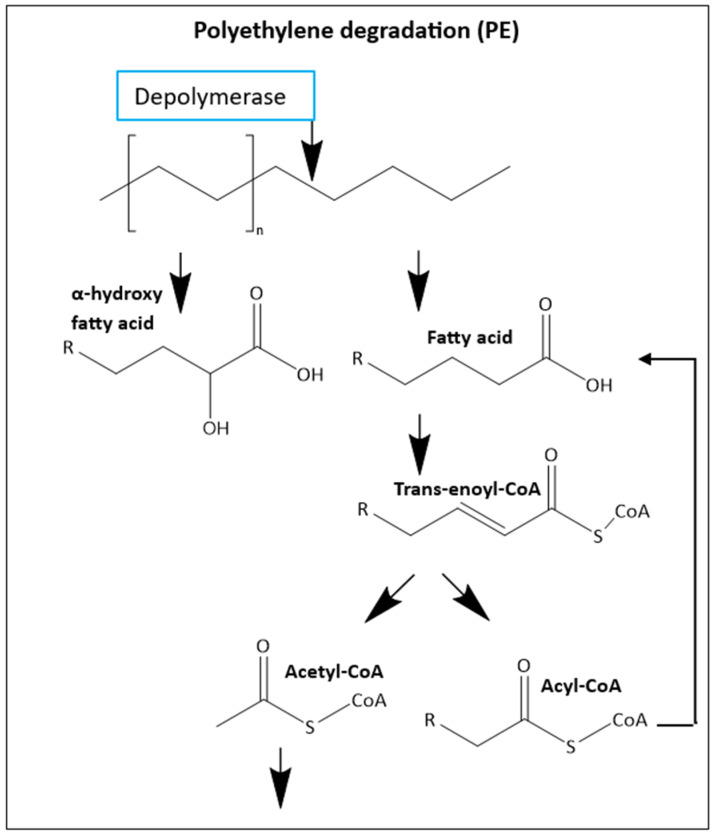
Proposed mechanism of PE degradation in *Tenebrio molitor* larvae (mealworms) presented by Zhong, Nong, Xie, Hui, and Chu [105].

**Figure 10 insects-16-00165-f010:**
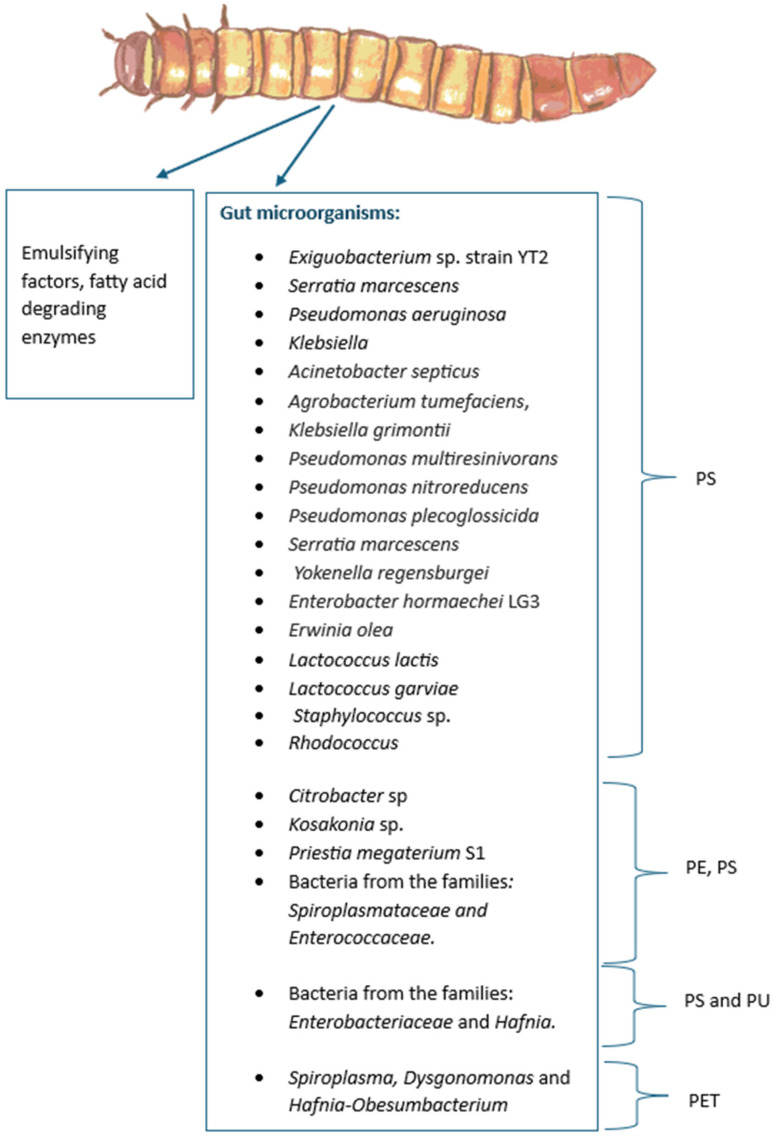
*Tenebrio molitor* uses a wide variety of gut microorganisms to degrade plastics.

**Figure 11 insects-16-00165-f011:**
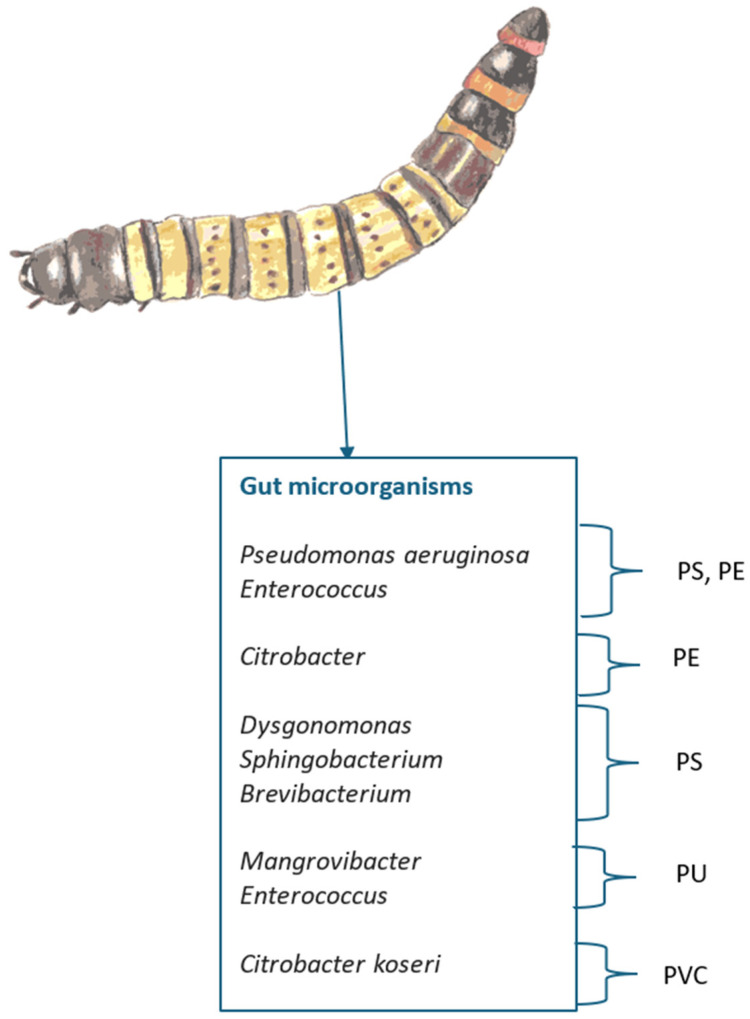
Zophobas can digest a wide variety of plastics with the aid of gut microbes.

**Figure 12 insects-16-00165-f012:**
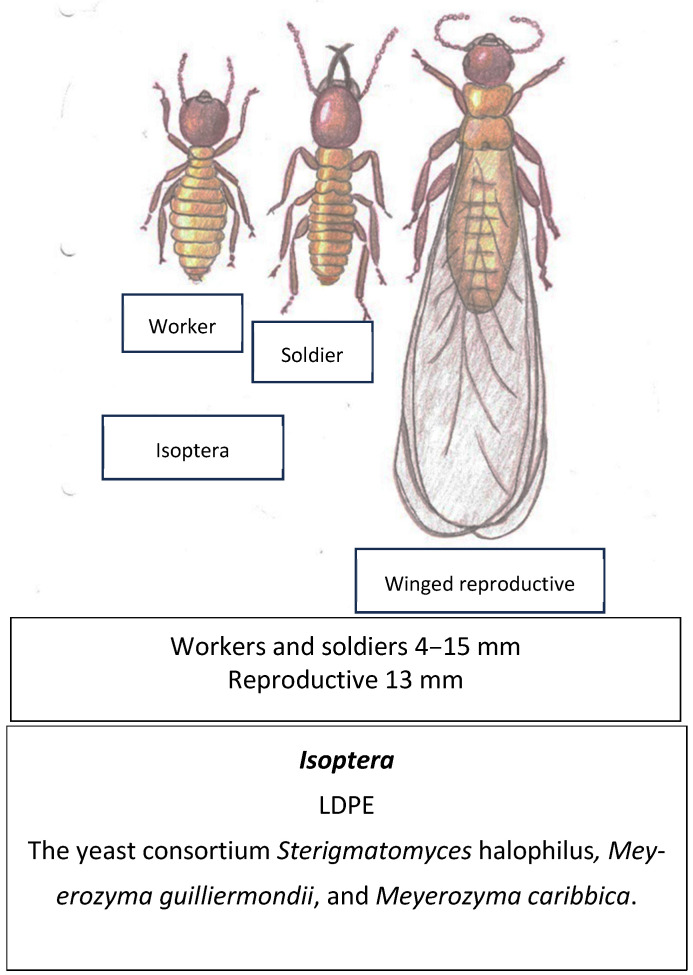
Termites live in colonies formed by workers, soldiers, and winged reproductive termites, which are represented in this figure [16].

## Data Availability

No new data were created or analyzed in this study.

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
