# Peer review of "Using Insect Larvae and Their Microbiota for Plastic Degradation"

_insects, 2025, doi:10.3390/insects16020165_

Round 1
Reviewer 1 Report
Comments and Suggestions for Authors
This review article describes major plastivore insects and possible enzymes and gut microbes responsible to plastic biodegradation. In my opinion, it can be published in Insects after major revision. I suggest following for manuscript improvement.
1) The authors should cover updated important plastivore insects. The manuscript does not specially report an important number of Tenebrio obscurus or dark mealworms, which are widely found in the USA, China and Western Europe (Peng et al., 2019). Biodegradation of plastics by T. obscurus has been reported by several investigators and summarized in recent review paper by Yang et al. (2024). The authors also do not include recent publications as on biodegradation of PS by the family of white grubs (Scarabaeidae: Coleoptera), i.e., Pretaetia Brevitarsis (soil-dwelling grub) (Jiang et al., 2024), and cockroaches (Blattodea:Blaberidae), i.e., Blaptica dubia (Li et al., 2024).
2) The Figures should be reorganized more professionally. I have suggestions for the following figures.
Figure 1. Add “2003-2009. Teenagers claim mealworms degrade PS at science fairs”.
Figure 2. Add estimated half-life of respective plastics by insects (e.g., mealworms, superworms).
Figure 3. It should include a pathway of abiotic followed by biotic reactions. In environment, it is common. Photo-aged plastics are further attacked by microbes in soil and aquatic environments.
Figures 5 and Figure 6 should be combined into one figure and also add the images of larvae because plastic degradation is performed by their larvae. The size range of moth and mature larva should also be indicated. For publication, the images should be looked professionally.
Figures 9, 10 and 13 should be combined as one figures plus images of mature larvae with the size of both larvae and adults. For publication, the images should be looked professionally. Dark mealworms should also be added.
3) Some special comments are described as bellow.
Line 17 and Line 52. In 2014, Yang et al. reported isolation of PE-degrading bacteria from PE-film-eating Plodia larvae but did not provide evidences of PE degradation by this insect. In 2015, Yang et al. provided first time evidences of PS biodegradation by Tenebrio molitor. Previously, high school students had hypotheses that mealworms degraded PS in earlier 2000s’ (See Yang et al., 2024, Front. Environ. Sci. Eng., 18(6), 78).
Line 32. Revised manuscript should update the literatures published in Dec. 2024.
Line 51-52. High school students claimed degradation of expanded polystyrene in science fair activities in 2000s should be added.
Line 101. Hydrolysable polymers PET and PU should have much shorter half-life in environment than non-hydrolysable plastics PE, PP, PVC and PS. Except for ref.22, you should find several review papers including estimated half-life of major plastics.
Line 210-217, Line 241-247. Recent research indicates that PET degrading enzymes can be selected directly form plastic-degrading mixed culture e.g., gut microbiome of mealworms rather isolated single or pure culture (see ES&T, 2024, 58, 17717-17731 ).
Line 339-354. The effectiveness of salivary enzymes have been questioned by some researchers (see Nature Communications, 2024, 15:8501) . Cite this paper and more further research work is needed to verify the enzymatic activities.
Line 420. Styrene has been proposed to be depolymerized product during PS degradation (Figure 8A). You should indicate that this pathway is still hypothesis since to date no report shows that styrene production has been verified during biodegradation by microbes or insects.
Line 622. Both cockroaches and termites belong to the order Blattodea. Cockroaches chewing and eating plastic film was reported in 1950s. Recently, biodegradation of PS by Dubia cockroaches has been confirmed. However, the ability of plastic degradation by termites has yet been verified using GPC, FTIR and mass balance analyses although some species are likely plastivore candidate. References 14, 165 and 166 do not provide direct evidences to confirm biodegradation of plastics by termites as reports on biodegradation of PE, PS and others by mealworms.
Line 690-710. This section on the analytical techniques should be rewritten. Comprehensive description and review on the analytic methods and test procedures of plastic degradation by insects can be found in Methods in Enzymology, 2021, 648, 95-129; Front. Environ. Sci. Eng., 2024,18 (6): 78. Mass balance is essential for the characterization of plastic degradation. GC/MC, HPLC/MS, and Py-GC/MC can be used to evaluate intermediates. Isotopic tracer (14C and 13C) and stable isotopic analysis (δ13C) are also useful tools.
Références
Jiang et al. et al. Soil-dwelling grub larvae of Protaetia brevitarsis biodegrade polystyrene: Responses of gut microbiome and host metabolism. Sci. Total Environ. 2024; 934: 173399.
Li et al. 2024.Cockroach Blaptica dubia biodegrades polystyrene plastics: Insights for superior ability, microbiome and host genes. J. Hazardous Materials. 2024, 479, 135756.
Mamtimin et al. 2024. Novel Feruloyl Esterase for the Degradation of Polyethylene Terephthalate (PET) Screened from the Gut Microbiome of Plastic-Degrading Mealworms (Tenebrio molitor Larvae).Environmental Science & Technology. 58, 17717-17731.
Yang et al. 2024. Radical innovation breakthroughs of biodegradation of plastics by insects: history, present and future perspectives. Front. Environ. Sci. Eng. 2024, 18(6): 78.
Reviewer 2 Report
Comments and Suggestions for Authors
Overall, the article makes a valuable contribution to the field of biodegradation and provides a review of the current state of research on insect larvae and their role in plastic degradation. With some minor revisions, it will be good for publication.
1. The title is informative and captures the essence of the research area. However, consider simplifying it to "Insect Larvae and Microbiota in Plastic Degradation" for brevity and impact.
2. The structure is logical, but the introduction could be more engaging. Consider starting with a statement on plastic pollution and its environmental implications. Each section could benefit from a brief summary or conclusion to reinforce the key points discussed.
3. Ensure consistent use of terminology throughout the document. For example, "plastic" and "polymer" are used interchangeably, which may cause confusion.
4. The article is rich in technical details, which are well-presented. However, for a broader audience, consider including a brief explanation of the “step-by-step” biochemical processes involved in plastic degradation by insects.
5. Improve the figures with more academic styles.
Round 2
Reviewer 1 Report
Comments and Suggestions for Authors
The revised manuscript has been improved significantly and can be accepted to be published.
Line 32 and Line 87: Replace "till June 2024" by "till September 2024".